# Serine catabolism is essential to maintain mitochondrial respiration in mammalian cells

Stephanie Lucas[1], Guohua Chen[1], Siddhesh Aras[2], Jian Wang[1,3]

**Breakdown of serine by the enzyme serine hydroxymethyltransferase (SHMT) produces glycine and one-carbon (1C) units. These serine catabolites provide important metabolic intermediates for the synthesis of nucleotides, as well as methyl groups for biosynthetic and regulatory methylation reactions. Recently, it has been shown that serine catabolism is required for efficient cellular respiration. Using CRISPR-Cas9 gene editing, we demonstrate that the mitochondrial SHMT enzyme, SHMT2, is essential to maintain cellular respiration, the main process through which mammalian cells acquire energy. We show that SHMT2 is required for the assembly of Complex I of the respiratory chain. Furthermore, supplementation of formate, a *bona fide* 1C donor, restores Complex I assembly in the absence of SHMT2. Thus, provision of 1C units by mitochondrial serine catabolism is critical for cellular respiration, at least in part by influencing the assembly of the respiratory apparatus.**

## Introduction

Apart from its role in protein synthesis, serine is a major metabolic source for generating one-carbon (1C) units in mammalian cells (de Koning et al, 2003). Two serine hydroxymethyltransferase (SHMT) enzymes, SHMT1 and 2, break down serine into glycine and methylene-tetrahydrofolate (THF) in the cytosol and mitochondria, respectively (Stover & Schirch, 1990; Stover et al, 1997). The latter serine catabolite feeds into cellular 1C pool, and either directly participates in thymidine synthesis or indirectly in purine or methionine synthesis after its oxidative or reductive conversion to formyl- or methyl-THF (Tibbetts & Appling, 2010). Because the 1C-derived products are key anabolic building blocks, sustaining the 1C pool is vital for cellular proliferation and is required for a number of physiological and pathophysiological processes ranging from stem cell renewal to cancer progression (Wang et al, 2009; Locasale, 2013). Consistent with their critical roles in supporting cell proliferation, SHMTs are highly active in many rapidly growing cancer cells and are important molecular targets for cancer intervention (Snell et al, 1988; Nikiforov et al, 2002; Ducker et al, 2017).

Interestingly, 1C metabolism also functionally interacts with mitochondrial oxidative phosphorylation (OXPHOS) system, the main process through which mammalian cells generate ATP. The OXPHOS system comprises an electron transport chain of four respiratory enzyme complexes (Complex I–IV) that use nutrient-derived redox potentials to drive Complex V (CV), the ATP synthase (Alberts et al, 2002). The protein components of the OXPHOS system are encoded by both nuclear and mitochondrial genes (Ott et al, 2016). It was recently shown that electron transport chain dysfunction owing to mitochondrial DNA (mtDNA) depletion dramatically alters the expression of SHMT2 as well as the production of 1C units from serine catabolism (Bao et al, 2016; Nikkanen et al, 2016). In addition, system-wide metabolic modeling indicates that oxidation of the serine-derived 1C units provides a significant fraction of the redox potential to drive ATP synthesis via OXPHOS (Vazquez et al, 2011; Tedeschi et al, 2013). These observations strongly suggest that the 1C metabolic cycle and the OXPHOS system are functionally coupled. Recent studies further demonstrated that serine catabolism by SHMT2 is required to maintain mitochondrial respiration in human cell lines (Minton et al, 2018; Morscher et al, 2018) and mouse tissues (Tani et al, 2018). Interestingly, these works revealed distinct mechanisms underlying a crucial role of SHMT2 in sustaining mitochondrial translation in different cell types (Minton et al, 2018; Morscher et al, 2018), indicating that complex mechanisms exist, linking serine catabolism to the modulation of the OXPHOS system.

In the present study, we independently investigated the metabolic adaptions in response to targeted deletion of SHMT enzymes in mammalian cells. Consistent with the previous reports (Minton et al, 2018; Morscher et al, 2018), we found that the cells lacking SHMT2, but not SHMT1, preferentially metabolized glucose to lactate and were unable to survive in the presence of galactose media, suggestive of mitochondrial dysfunction in the absence of SHMT2. Mechanistically, we found that SHMT2 is dispensable for mtDNA maintenance and OXPHOS gene expression. However, our results strongly suggest that SHMT2 plays a critical role in supporting the assembly of Complex I by supplying the 1C intermediate derived

[1]Department of Pathology, Wayne State University, Detroit, MI, USA  [2]Center for Molecular Medicine and Genetics, Wayne State University, Detroit, MI, USA  [3]Cardiovascular Research Institute, Wayne State University, Detroit, MI, USA

Correspondence: jianwang@med.wayne.edu

from serine catabolism. Together, our findings revealed a novel regulatory link between SHMT2-mediated 1C metabolism and the maintenance of the mitochondrial respiratory chain in mammalian cells.

## Results

### Loss of SHMT2 stimulates aerobic glycolysis

To examine the function of SHMT1 and SHMT2 enzymes, each gene was independently ablated in 293A cells using CRISPR-Cas9 technology. Two mutant cell lines for each *SHMT* gene were generated using two different small-guide RNAs (sgRNAs) to target each *SHMT* gene at different genomic locations. Transfections of Cas9 alone into the human embryonic kidney 293A (HEK293) cells did not affect SHMT expression and were used as the WT control. Both SHMT1 (Δ*SHMT1*) mutant lines and both SHMT2 (Δ*SHMT2*) mutant lines, created by introduction of the gRNAs, showed complete loss of the respective protein, as determined by the Western blotting analysis (Fig 1A).

Consistent with a previous report (Ducker et al, 2016), both Δ*SHMT1* and Δ*SHMT2* cells grew similarly to the WT when maintained in the standard DMEM media, which contains high levels of glucose (Fig 1B). This may reflect a redundancy of SHMT paralogs to supply 1C units for supporting cell proliferation (Ducker et al, 2016). Interestingly, it was noted that Δ*SHMT2* cells consistently turned the culture media acidic faster than the WT or Δ*SHMT1* cells,

suggesting an elevated rate of glycolysis as a result of SHMT2 loss. This possibility was supported by the observation that Δ*SHMT2* cells had significantly increased production of lactic acid, the end product of aerobic glycolysis, whereas Δ*SHMT1* cells did not (Fig 1C). In addition, the levels of key glycolytic enzymes, including pyruvate kinase M (PKM) and lactate dehydrogenase A, were modestly elevated in the Δ*SHMT2* cells (Fig 1A). Thus, these results indicate that the cells lacking SHMT2 prefer partial metabolism of glucose to lactate.

### SHMT2 is essential for maintaining mitochondrial respiration

Up-regulation of aerobic glycolysis can indicate a metabolic adaption to mitochondrial dysfunction (Wu et al, 2007). To test this possibility, we forced the cells to rely on mitochondrial respiration by growing the cells in media containing galactose instead of glucose (Robinson et al, 1992). Under these conditions, we observed that the Δ*SHMT2*, but not the Δ*SHMT1*, cells exhibited reduced intracellular ATP and were unable to proliferate (Fig 2A and B), indicating mitochondrial dysfunction in the absence of SHMT2. In contrast, the Δ*SHMT2* cells were able to maintain a normal intracellular ATP level in media containing glucose (Fig 2C). However, we observed a significant reduction of basal oxygen consumption by the Δ*SHMT2* cells when maintained in standard glucose media (Fig 2D). Together, these results demonstrate that the mitochondrial serine catabolic enzyme SHMT2 is essential for maintaining normal levels of cellular respiration.

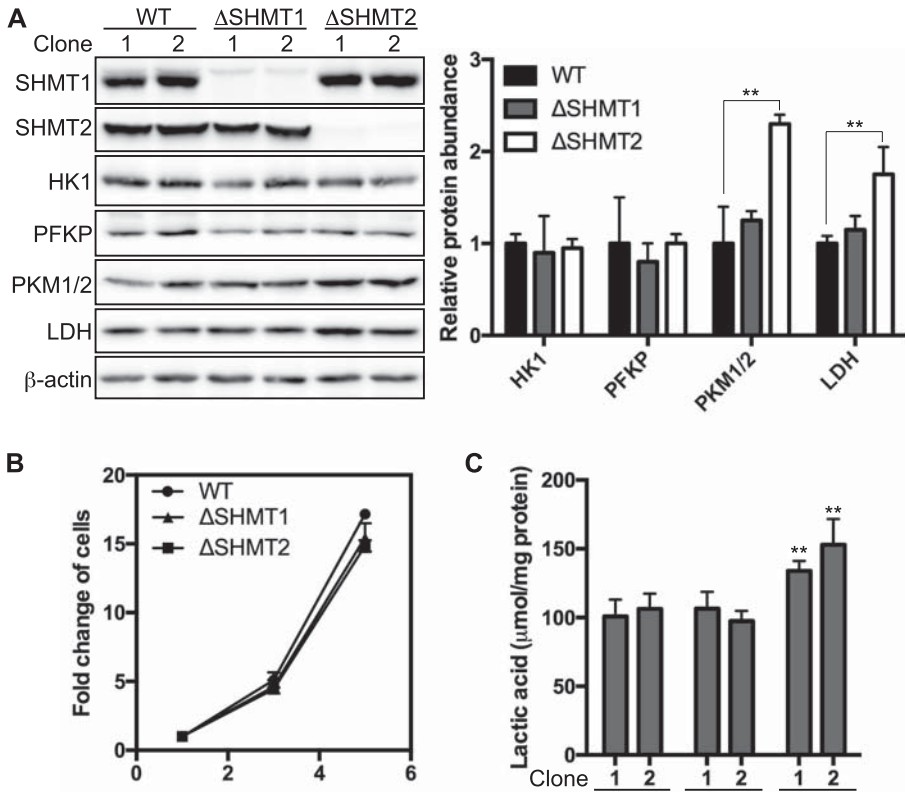

**Figure 1.  Effects on glycolysis by the deletion of serine catabolic enzymes in 293A cells.**
**(A)** Protein levels of SHMT1, SHMT2, hexokinase 1 (HK1), phosphofrutokinase (platelet type, PFKP), pyruvate kinase (muscle type ½, PFKM1/2), lactate dehydrogenase A (LDHA), and β-actin were measured in the WT, Δ*SHMT1*, or Δ*SHMT2* 293A cells by Western blotting. Two independent cell clones of each genotype were examined. Densitometry quantification of the glycolytic proteins followed by normalization to β-actin is plotted on the right. **P < 0.01 (t test). Data are presented as mean ± SD for three independent experiments. **(B)** Proliferation rates of the WT, Δ*SHMT1*, and Δ*SHMT2* 293A cells in the DMEM with 4.5 g/liter glucose. Data are presented as mean ± SD (n = 5). **(C)** Measurement of the lactic acid production from the WT, Δ*SHMT1*, and Δ*SHMT2* 293A cells that were grown in DMEM for 48 h. Data are presented as mean ± SD (n = 4) **P < 0.01 (t test).

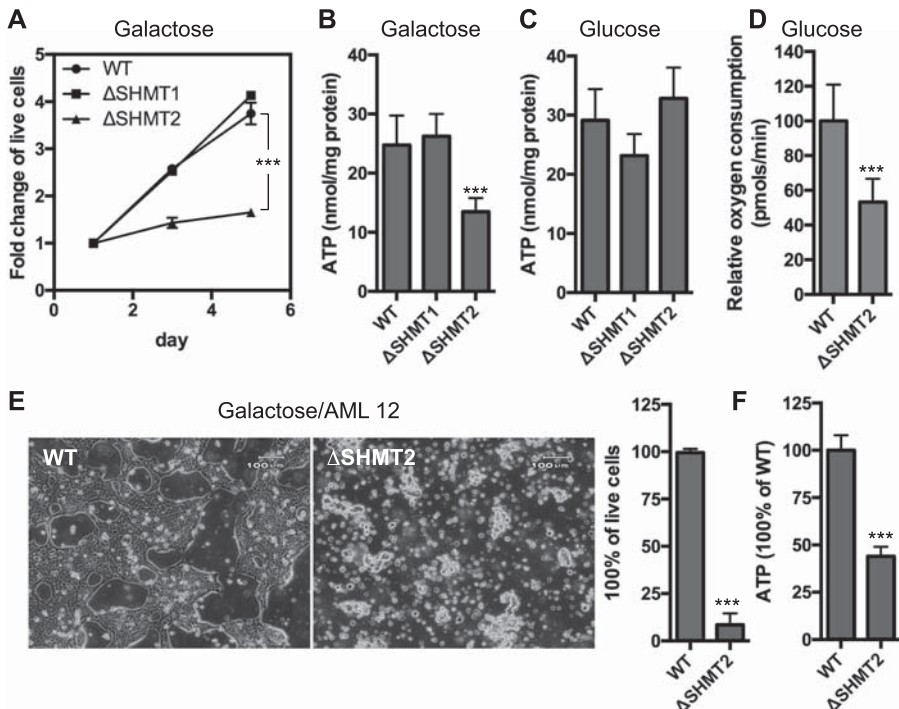

**Figure 2. Effects on mitochondrial respiration by the deletion of serine catabolic enzymes in 293A and AML12 cells.**
**(A)** Measurement of the cell proliferation of the WT, ΔSHMT1, and ΔSHMT2 293A cells in the DMEM-based galactose media. **(B, C)** Measurement of the intracellular ATP levels in the WT, ΔSHMT1, and ΔSHMT2 293A cells that were grown in the galactose (B) or glucose (C) media for 24 h. **(D)** Measurement of the basal oxygen consumption rates of the WT and ΔSHMT2 293A cells that were grown in the glucose media. **(E)** Phase-contrast images illustrating the WT and ΔSHMT2 AML12 cells that were grown in the galactose media for 72 h. The percentages of the live cells were plotted on the right. **(F)** Measurement of the intracellular ATP levels of the WT and ΔSHMT2 AML12 cells that were grown in the galactose media for 72 h. ***$P < 0.001$ ($t$ test). Data are presented as mean ± SD. n = 5 for (A); n = 4 for (B–E); and n = 6 for (F).

To test whether the dependence of cellular respiration on SHMT2 extends to another cell type, we deleted *SHMT2* in the AML12 mouse hepatocyte using CRISPR-Cas9 genome editing system. Consistent with the observations in 293A cells, knockout of *SHMT2* in AML12 cells led to cell death (Fig 2E) and a decrease in intracellular ATP levels (Fig 2F) in the presence of galactose media. Thus, the essential role for SHMT2 to maintain respiration is conserved between two different cell types from different species (human and mouse).

### SHMT2 is dispensable for the maintenance of mtDNA content and OXPHOS gene expression

To test whether loss of *SHMT2* affects the expression of the individual OXPHOS enzyme complexes, we subjected the WT and mutant 293A cells to Western blotting analysis with Total OXPHOS antibody cocktail. Interestingly, we observed that knockout of *SHMT2* selectively decreased the level of Complex I as revealed by its marker NADH:ubiquinone oxidoreductase subunit B8 (Fig 3A). Moreover, such decrease in Complex I level in the ΔSHMT2 cells was further confirmed by measuring NADH:ubiquinone oxidoreductase core subunit S1 (NDUFS1), a core subunit of Complex I (Fig 3A). These results indicate that cells lacking *SHMT2* have a defect in the mitochondrial OXPHOS system, which is particularly associated with the Complex I.

The mammalian mitochondrial genome encodes 13 peptides that are core components of the OXPHOS system (Neupert, 2016). Loss of SHMT2 reduces cellular deoxynucleotide pool (Ducker et al, 2016), which can be associated with mtDNA depletion (Bourdon et al, 2007; Suomalainen & Battersby, 2018). However, when we measured mtDNA copy numbers, we observed that the ΔSHMT2 cells had similar mtDNA content to the WT and ΔSHMT1 cells (Fig 3B). Next, we determined the relative mRNA expression of the OXPHOS system genes in the ΔSHMT2 cells to the WT, by next-generation

sequencing (for the nuclear-encoded transcripts) and by quantitative RT-PCR (qRT-PCR) analysis (for the mitochondria-encoded transcripts). The results showed that knockout of *SHMT2* did not reduce the expression of any mRNA species that encodes the OXPHOS enzyme (Fig 3C and D). The cellular 1C pool connects to the synthesis of formyl-methionyl-tRNA, a metabolite that is required for the initiation of mitochondrial translation (Kozak, 1983). This raises the possibility that loss of SHMT2 might reduce the synthesis of the mitochondria-encoded proteins. However, when we measured mitochondrial protein synthesis by metabolically labeling the cells with [35]S-methione/cysteine in the presence of the cytosolic protein translation inhibitor emetine, we observed that the ability of the ΔSHMT2 cells to synthesize the mitochondria-encoded proteins was comparable with the WT (Fig 3E). This is in contrast to the recent observations that deletion of *SHMT2* negatively impacted mitochondrial translation in the Jurket and HCT116 cells (Minton et al, 2018; Morscher et al, 2018). Although a possible negative impact by *SHMT2* deletion on mitochondrial translation cannot be entirely excluded in our cell model, our results suggest that the regulation of the OXPHOS system by SHMT2 may also occur after the synthesis of mitochondria-encoded OXPHOS proteins.

### Loss of SHMT2 severely reduces the level of mature Complex I

We next tested whether deletion of *SHMT2* influences the expression of the individual mature OXPHOS complexes. Blue native gel electrophoresis (BNGE) allows for the separation of multiprotein complexes in a native conformation with high resolution (Schägger & von Jagow, 1991). Using BNGE to examine the steady-state levels of the mature OXPHOS complexes in the WT and ΔSHMT2 293A cells, we observed that loss of SHMT2 essentially abolished the expression of the monomeric Complex I and the Complex III dimer (CIII₂)/CIV supercomplex (Fig 4A). In contrast, the levels of the CIII₂ and CV

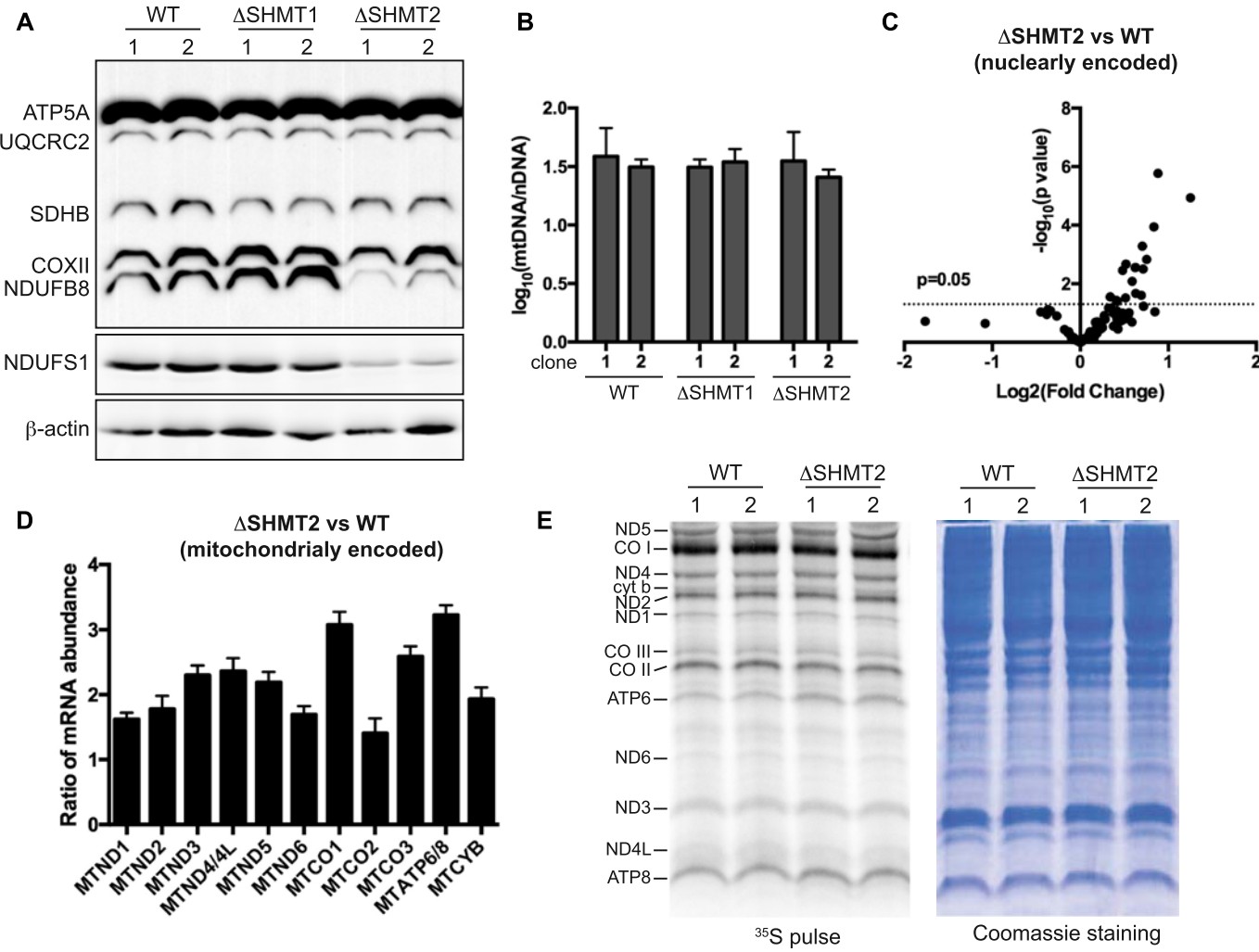

**Figure 3.  Effects on the content of mtDNA and the expression of the OXPHOS genes by deletion of SHMT2 in 293A cells.**
**(A)** Measurement of the levels of the respiratory chain complexes (RCCs) in the WT, Δ*SHMT1*, and Δ*SHMT2* 293A cells by Western blotting, using the Total OXPHOS antibody cocktail to simultaneously detect the representative components of each individual RCCs: CI-NADH:ubiquinone oxidoreductase subunit B8; CII-succinate dehydrogenase complex subunit B; CIII-ubiquinol-cytochrome C reductase core protein 2; CIV-cytochrome C oxidase II; and CV-ATP synthase F1 subunit alpha. NDUFS1, an additional CI marker, and β-actin, the loading control, were also measured. **(B)** Bar graph illustrating the mtDNA copy numbers of the WT, Δ*SHMT1*, and Δ*SHMT2* 293A cells. Data are presented as mean ± SD (n = 4). **(C)** Scatter plot illustrating the log2-transformed relative mRNA levels of the nuclear-encoded OXPHOS genes in the Δ*SHMT2* 293A cells to the WT control, as determined by next-generation sequencing; $t$ test (n = 3). **(D)** Bar graph illustrating the relative mRNA levels of the mitochondria-encoded OXPHOS genes in the Δ*SHMT2* 293A cells to the WT control, as determined by qRT-PCR analysis. Data are presented as mean ± SD (n = 3). **(E)** Metabolic radiolabeling to determine the synthetic rates of the mitochondria-encoded proteins in the WT and Δ*SHMT2* 293A cells. Left, autoradiography to visualize the mitochondria-encoded OXPHOS proteins; right, Coomassie blue staining of total cellular proteins.

monomer were unaffected in the Δ*SHMT2* cells (Fig 4A). The dramatically decreased Complex I level and activity were further revealed in the Δ*SHMT2* cells using a Western blotting analysis probed with NDUFS3 antibody (Fig 4B) and an in-gel NADH oxidation assay (Fig 4C), respectively. In conjunction with the analyses on the levels of the monomeric CII and CIV (data shown below), we conclude that loss of SHMT2 selectively impedes the assembly of mature Complex I and the CIII₂/CIV supercomplex in 293A cells.

### Reconstitution of SHMT2 restores Complex I in the SHMT2-deficient cells

To firmly establish the causative relationship between SHMT2 expression and Complex I functionality, we re-expressed SHMT2 into

the Δ*SHMT2* 293A cells. The results showed that exogenous expression of SHMT2 protein in the Δ*SHMT2* cells to ~70% of the WT level led to significant recovery of Complex I level (Fig S1A) and activity (Fig S1B), as well as improvement of cell survival in galactose media (Fig S1C). Thus, genetic restoration strongly supports the conclusion that SHMT2 is required for Complex I assembly.

### Formate supplementation restores Complex I in the SHMT2-deficient cells

Loss of SHMT2 leads to glycine auxotrophy (Pfendner & Pizer, 1980) and depletion of formyl-THF (Ducker et al, 2016). To test whether serine catabolites are involved in the regulation of Complex I, we supplemented the Δ*SHMT2* 293A cells with the increasing amounts

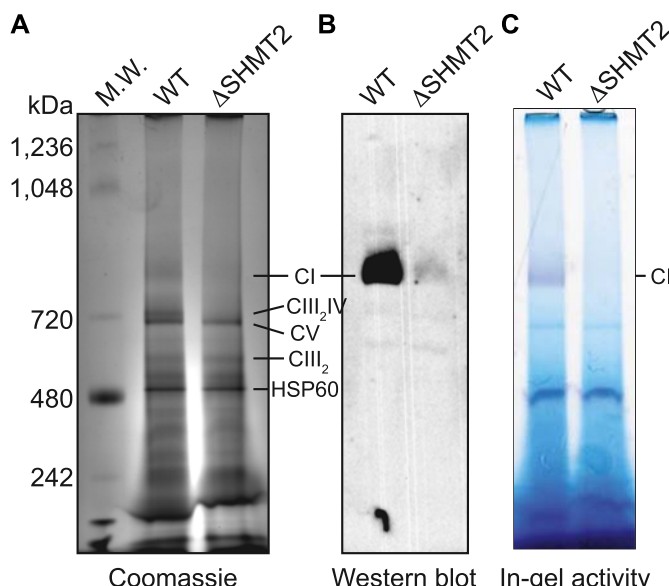

**Figure 4. Effects on the assembly of mature respiratory chain complexes by the deletion of SHMT2.**
**(A–C)** BNGE to detect individual mature respiratory complexes in the mitochondrial preparation from the WT and ΔSHMT2 293A cells, followed by visualization with Coomassie blue staining (A), Western blotting with the antibody against a Complex I (CI) marker NDUFS3 (B), or Complex I in-gel activity assay (C). CI, Complex I; CIII$_2$/IV, CIII$_2$/IV supercomplex; HSP60, heat shock protein 60.

of glycine or formate, a *bona fide* 1C donor (Brosnan & Brosnan, 2016). We observed that formate, but not glycine, significantly increased the levels of NDUFS1, a core subunit of Complex I, in a time- and dose-dependent manner (Fig 5A and B). We next determined how formate treatment influences the expression of the individual mature OXPHOS complexes. Using BNGE analyses, we demonstrated that loss of SHMT2 selectively abolished the expression of mature Complex I without affecting that of the other four individual complexes, including CII, CIII$_2$, CIV, and CV (Fig 5C, lane 1 versus 3). Importantly, formate treatment completely restored Complex I assembly (Fig 5C) and activity (Fig 5D) in ΔSHMT2 cells, as well as the proliferation of ΔSHMT2 cells in galactose (Fig 5E). These results strongly suggest that the effect of SHMT2 loss on Complex I assembly is mediated by altered mitochondrial 1C metabolism.

## Discussion

Serine is uniquely positioned at a metabolic crossroad that connects the major catabolic and anabolic pathways (de Koning et al, 2003; Kalhan & Hanson, 2012). Although synthesized from a glycolytic catabolite, it can be broken down to supply key anabolic intermediates for the synthesis of a number of important biomolecules, including nucleotides, lipids, and glutathione. It is not surprising, therefore, that serine itself is a key regulator of central metabolic pathways. This role is well illustrated by the allosteric regulation of serine on PKM2—a key glycolytic enzyme whose role is to partition glucose carbon into biosynthetic pathways in cancer cells (Chaneton et al, 2012; Ye et al, 2012). Here, we find that catabolism of serine by

SHMT2 is a regulatory determinant of cellular respiration. Moreover, we further demonstrate that the 1C units produced by SHMT2-mediated serine catabolism are essential for the assembly of functional Complex I. Our investigations, thus, uncovered a novel metabolic checkpoint for the assembly of the respiratory apparatus, suggesting that transformation of serine into 1C units coordinates the bioenergetic and biosynthetic pathways in mammalian cells.

Cellular adaptions to the loss of the serine-catabolic pathway either in the cytosol or mitochondria were investigated in the present study. We observed that loss of the SHMT enzyme in either cellular compartment posed minor impact on cell proliferation when the cells were grown in complete media. This is consistent with previous reports that SHMT paralogs are redundant for supply of 1C units in support of cell proliferation (Bao et al, 2016; Ducker et al, 2016). Interestingly, we observed that loss of SHMT2, but not SHMT1, led to elevation of aerobic glycolysis, indicating the involvement of SHMT2 in retrograde regulation on glycolysis in the cytosol. However, the up-regulation of glycolysis in response to SHMT2 deficiency cannot be easily explained by the allosteric activation of PKM2 by serine (Chaneton et al, 2012; Ye et al, 2012) because cellular serine levels were shown to be unaltered in the HEK293 cells lacking SHMT2 in previous studies (Bao et al, 2016; Ducker et al, 2016). Instead, our findings suggest that such elevation of glycolysis is a compensatory response to mitochondrial dysfunction. This compensation allows the substrate-level phosphorylation of glycolysis to substitute energy production from mitochondrial respiration, as previously observed in some cancer cells (Wu et al, 2007). Our observation made in 293A cells is consistent with the previous observations that SHMT2, but not SHMT1, is essential for maintaining mitochondrial respiration in the Jurket and HCT116 cells (Minton et al, 2018; Morscher et al, 2018). The results of the present study support the conclusion that SHMT2 is a critical metabolic checkpoint for OXPHOS function in a variety of mammalian cell types.

As a major metabolic route to produce 1C units, serine catabolism by SHMT2 might influence the OXPHOS gene expression in several ways including the influence on (i) cellular nucleotide homeostasis (Bao et al, 2016; Ducker et al, 2016) and therefore mtDNA replication (Anderson et al, 2011); (ii) mRNA expression of the OXPHOS genes (Mentch et al, 2015; Kottakis et al, 2016); and (iii) biosynthesis of the mitochondria-encoded OXPHOS proteins (Minton et al, 2018; Morscher et al, 2018). Recent studies demonstrated that SHMT2 is necessary for sustaining mitochondrial translation in the Jurket human leukemic (Minton et al, 2018) and HCT116 human colon cancer (Morscher et al, 2018) cells. Interestingly, these studies revealed distinct mechanisms that account for different types of translational abnormalities consequent to *SHMT2* deletion in these two cell types. In response to *SHMT2* deletion in Jurket cells, Minton et al (2018) found a decrease in formyl-methionyl-tRNA synthesis. This resulted in impaired mitochondrial translation initiation, leading to defective global OXPHOS protein synthesis in mitochondria (Minton et al, 2018). Morscher et al (2018), on the other hand, found that *SHMT2* deletion in HCT116 cells caused a reduction in the methylation of certain tRNA species. This resulted in ribosomal stalling in HCT116 cells, leading to defective translation for a subset of the mitochondria-encoded

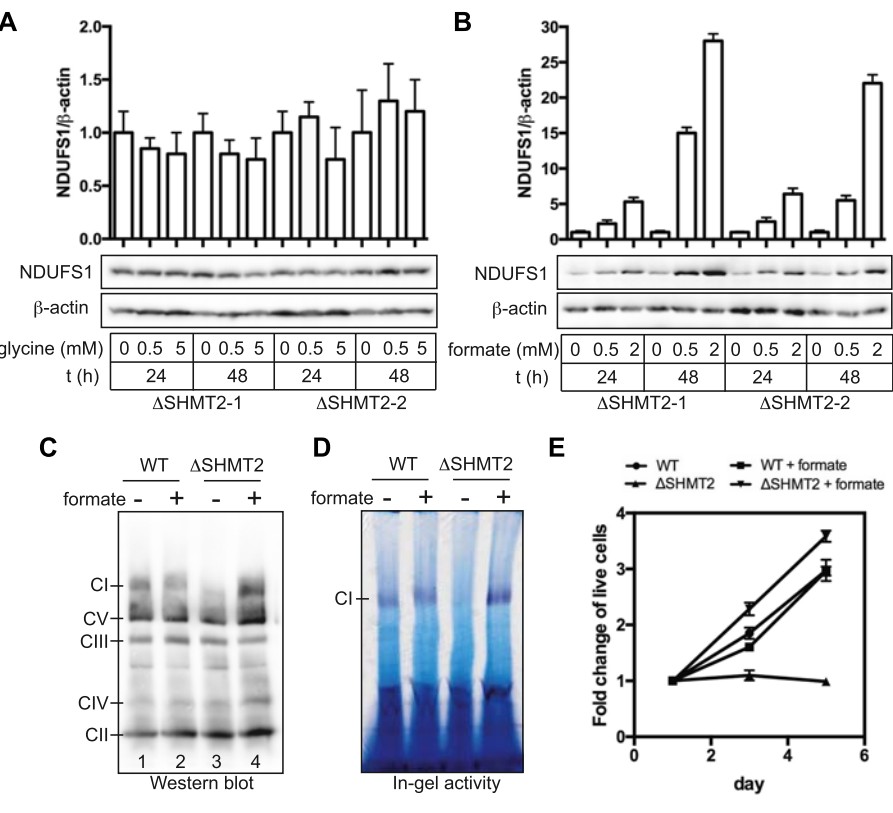

**Figure 5. Effects of glycine and formate on the assembly of Complex I.**
**(A, B)** Time- and dose-dependent effects on the levels of Complex I core subunit NDUFS1 by the supplementation of the ΔSHMT2 293A cells with glycine (A) or formate (B). Densitometry quantification of NDUFS1 followed by normalization to β-actin is plotted on the top. Data are presented as mean ± SD for three independent experiments. **(C, D)** BNGE resolution of the mitochondrial preparation from the WT and ΔSHMT2 293A cells that were treated with or without 2 mM formate for 72 h, followed by visualization of the individual mature respiratory chain complexes by the Western blotting probed with the Total OXPHOS antibody cocktail (C) and the Complex I (CI) in-gel activity assay (D). CI, Complex I. **(E)** Proliferation rates of the WT and ΔSHMT2 293A cells in the galactose media supplemented with or without 2 mM formate. Data are presented as mean ± SD (n = 4).

OXPHOS proteins (Morscher et al, 2018). Consistent with these previous reports, our results showed that deletion of *SHMT2* did not decrease the mtDNA content or the expression of the mRNAs that encode OXPHOS enzymes. Surprisingly, using $^{35}$S-methionine/cysteine pulse labeling, we were unable to observe an appreciable alteration in the biosynthesis of the mitochondria-encoded OXPHOS proteins in the SHMT2-deficient 293A cells. Given that differential translational responses to *SHMT2* deletion were also observed from the different cell types in the previous studies (Minton et al, 2018; Morscher et al, 2018), it is plausible that SHMT2 could modulate mitochondrial translation in a cell-type–specific manner. Nevertheless, we cannot entirely exclude the possibility that loss of SHMT2 could pose a negative impact on the mitochondrial translation in 293A cells. A more sophisticated approach such as ribosome profiling (Ingolia, 2016; Morscher et al, 2018) is desired to further investigate this matter in 293A cells. Interestingly, we further found that deletion of *SHMT2* selectively and dramatically reduced the steady-state levels of the mature Complex I and the CIII$_2$/CIV supercomplex, and that exogenous 1C supplementation restored the assembly of Complex I in the SHMT2-deficient 293A cells. These results raise the novel possibility that the 1C metabolic intermediate derived from mitochondrial serine catabolism is a metabolic determinant for the assembly of respiratory chain complex, in addition to its contribution to mitochondrial translation.

Assembly of the respiratory chain determines the competence of the OXPHOS system. However, the molecular mechanism for this process is far from understood (Stroud et al, 2016; Guerrero-Castillo et al, 2017). Although our findings demonstrate that SHMT2-mediated 1C metabolism is required for Complex I assembly, the exact mechanism by which this occurs is unknown and requires further investigation. Interestingly, it has been recently shown that ribosome stalling mostly impaired Complex I activity in the absence of SHMT2 (Morscher et al, 2018). This raises the question of whether lack of mitochondrial serine catabolism influences potential functional interaction between mitochondrial translation and complex assembly. However, it could also be speculated that the metabolites derived from 1C pool may convey a regulatory signal to molecular factor(s) involved in complex assembly. As such, some Complex I assembly factors, for example, the NADH:ubiquinone oxidoreductase complex assembly factor 5 and the NADH:ubiquinone oxidoreductase subunit A9 were found to harbor *S*-adenosylmethionine and NADPH (Rhein et al, 2013; Fiedorczuk et al, 2016), two important downstream metabolites derived from the 1C cycle (Tibbetts & Appling, 2010; Ducker & Rabinowitz, 2017). It is, thus, interesting to further test whether these factors might influence Complex I assembly in response to the fluctuations of cellular 1C pool. In addition, the results presented here provoke questions beyond the mechanistic aspects of how 1C metabolites are involved in Complex I assembly. For example, does the fluctuation of the 1C pool dynamically regulate Complex I assembly in a temporal–spatial manner? Is this regulation also important for the maintenance of metabolic homeostasis at a systemic level? As such, the findings made here may also suggest that targeting serine catabolism could be useful for intervening in a broad spectrum of the human pathologies associated with mitochondrial dysfunction, in addition to cancer.

# Materials and Methods

## Cell culture, plasmid construction, and mutant cell line establishment

HEK293A (Invitrogen) and 293T (American Type Culture Collection) cells were maintained in DMEM medium supplemented with 10% FBS, 100 U of penicillin/ml, and 0.1 ng of streptomycin/ml. Murine AML12 hepatocytes (American Type Culture Collection) were maintained in a 1:1 mixture of DMEM and Ham's F12 medium, supplemented with 10% FBS, 1:100 insulin-transferrin-selenium (Invitrogen), 100 U of penicillin/ml, and 0.1 ng of streptomycin/ml.

For construction of the targeting vector against human SHMT, the sgRNA sequences were determined using a web bioinformatics tool (http://crispr.mit.edu) (Ran et al, 2013). The oligo DNAs that encode sgRNA were annealed and cloned into pSpCas9(BB)-2A-Bsd, a modified vector from pSpCas9(BB)-2A-Puro (PX459; Addgene) made by replacing the puromycin-resistant to a blasticidin-resistant gene. For construction of the targeting vector against murine SHMT2, the sgRNA sequences were determined using a web bioinformatics tool (https://benchling.com/academic). The oligo DNAs that encode the sgRNA were annealed and cloned into an AAV targeting vector (PX602; Addgene). The sgRNA sequences used are listed in Table S1.

For establishment of the 293A cell line with targeted deletion of SHMT, the targeting vector was transfected into cells using lipofectamine 2000 (Invitrogen). The drug-resistant cell clones were obtained by selecting the transfectants with 5 μg/ml of blasticidin and then were screened for loss of SHMT protein expression by Western blotting. For establishment of the AML12 cell line with targeted deletion of SHMT2, the adeno-associated virus serotype DJ/8 viral particles that harbor the CRISPR-Cas9 system were produced using the AAV Helper Free Packaging System (Cell Biolabs) and were then applied to the cells. The infectants were seeded into 96-well plates at the single-cell level using a flow cytometer. The cell clones were expanded and then screened for loss of SHMT2 protein expression by Western blotting. The genomic modifications were confirmed by automatic DNA sequencing.

For rescue of SHMT2 expression in the 293A mutant cell line, the full-length human SHMT2 cDNA was cloned into pcDNA3.1.puro, a modified vector of pcDNA3.1 (Invitrogen) made by replacing the G418-resistant with a puromycin-resistant gene. The synonymous mutations at the sgRNA binding site were introduced into the expression vector using a site-directed mutagenesis kit (Agilent) to evade gene targeting. The resultant SHMT2 expression vector was transfected into the SHMT2-knockout cells, and the drug-resistant clones were obtained by selection with 2 μg/ml of puromycin and were then screened for gain of SHMT2 protein expression by Western blotting.

## Western blotting

Cells were washed twice in PBS and lysed in cytoplasmic lysis buffer (25 mM Tris–HCl, pH 7.5, 40 mM NaCl, and 1% Triton X-100). Protein concentrations were determined with the Bradford reagent (Bio-Rad). Cell lysates (40 μg) were resolved by SDS-PAGE, and proteins were transferred onto nitrocellulose filters. The blots were saturated with 5% nonfat milk and probed with antibodies against SHMT1 (#HPA023314; Sigma-Aldrich, 1:1000), SHMT2 (#HPA020543; Sigma-Aldrich, 1:1000), HK1 (#2024; Cell Signaling Technology, 1:1000), lactate dehydrogenase A (#3582; Cell Signaling Technology, 1:1000), PFKP (#8164; Cell Signaling Technology, 1:1000), PKM1/2 (#3190; Cell Signaling Technology, 1:1000), NDUFS1 (sc-271510; Santa Cruz, 1:1000), NDUFS3 (sc-374282; Santa Cruz, 1:1000), β-actin (A2066; Sigma-Aldrich, 1:1000), or Total OXPHOS Human Antibody Cocktail (ab110411; Abcam, 1:1000). Following a wash with phosphate buffered saline with 0.1% Tween 20, the blots were incubated with peroxidase-coupled goat anti-rabbit immunoglobulin G (Sigma-Aldrich, 1:5000). The immunolabeled protein bands were detected by enhanced chemiluminescence (ECL) method (Perkin Elmer). Densitometric analysis of the blots was performed using Image Quant TL software (GE Healthcare).

## Determination of cell proliferation and cell survival

Cells were seeded on 6-cm plates and grown for the indicated time intervals in the glucose or galactose media. The dead cells were excluded with trypan blue staining (Invitrogen) and the number of live cells was measured using a hemocytometer under a light microscope.

## Measurements of lactic acid, ATP, and basal oxygen consumption

Measurement of lactic acid was adapted from Brandt et al (1980). In brief, appropriate amount of culture media was incubated at RT for 30 min in a final 100 μl of reaction mix containing 160 mM Tris–hydrazine, pH 9.0, 2.5 mM NAD$^+$, 0.01% BSA, and 8 U lactate dehydrogenase. The amount of lactic acid was extrapolated from a standard curve based on the Ab340 reading recorded on a microplate reader (BMG LABTECH). Cellular ATP level was determined using ENLITEN ATP assay kit (Promega) according to the manufacturer's specification. Intact cellular oxygen consumption was measured in the WT and ΔSHMT2 cells on an XFe24 seahorse bioanalyzer (Agilent) on plating 5 × 10$^4$ cells per well. Data have been represented as oxygen consumption relative to the WT cells.

## Determination of mRNA expression by next-generation sequencing and qRT-PCR

Total cellular RNA was isolated using the Trizol reagent (Invitrogen). To determine the mRNA expression of the nuclear-encoded OXPHOS genes, cDNA libraries compatible for Illumina sequencing were prepared by using the QuantSeq 3' mRNA-seq Reverse (REV) Library Prep Kit (Lexogen) according to the manufacturer's instruction. The resultant cDNA libraries were assessed using a TapeStation (Agilent) and subjected to 100-bp single-end sequencing using the Illumina HiSeq 2500 system at the Wayne State University Applied Genomics Technology Center. Raw sequencing reads in FASTQ format were processed with Trimmomatic (Bolger et al, 2014) to remove low-quality and unknown sequences. To quantify transcript abundance, the processed reads in FASTA format were mapped to the hg19 human reference genome using bowtie2 (Langmead & Salzberg, 2012). Transcript abundance in count per million was determined using eXpress (Roberts & Pachter, 2013), and differential gene expression was determined using edgeR (Robinson et al, 2010).

To determine the mRNA expression of the mitochondria-encoded OXPHOS genes, cDNA libraries were constructed by random priming using the SuperScript III First-Strand Synthesis System (Invitrogen). The synthesized cDNA was used as a template for qRT-PCR with SYBR Green qPCR Master Mixes (Thermo Scientific) on an Mx3000P cycler (Stratagene). All mRNA levels were determined as the delta–delta threshold cycle ($\Delta\Delta C_T$) and normalized to peptidylprolyl isomerase A mRNA level. The PCR primers used are listed in Table S2.

### Determination of mtDNA copy number

mtDNA copy number was determined by calculating the ratio of the mitochondria-encoded ND1 gene levels to the nuclear-encoded 28S rRNA gene levels using qPCR analysis. To quantify each gene level, serial dilutions of the cloned gene-specific PCR fragments were used to create a standard curve. The exact gene copy numbers of the ND2 and 28S rRNA genes were obtained by plotting the log-transformed $\Delta\Delta C_T$ values against the standard curves. The PCR primers used are listed in Table S2.

### Measurement of biosynthesis of the mitochondria-encoded proteins

Cells growing on 6-cm plates to 90% confluency were metabolically labeled with 400 µCi EastTag EXPRESS35S Protein Labeling Mix (PerkinElmer) in the presence of 100 µg/ml emetine for 1 h. Cell pellets were suspended in 1× Laemmli sample buffer and lysed by boiling for 10 min. Equal amounts of the extracts were resolved on a 16.5% tricine gel and dried on a GelAir gel dryer (Bio-Rad). The radiolabeled proteins were visualized by a phosphorimager (Molecular Dynamics).

### Mitochondria isolation and BNGE

To isolate mitochondria, cell suspensions were incubated in ice-cold homogenization buffer (10 mM Tris–HCl, pH 7.5, 250 mM sucrose, and 1 mM EDTA) for 15 min and then homogenized with a glass douncer by 15 strokes. After centrifugation at 600 $g$, 4°C for 10 min, the supernatants were further centrifuged at 11,000 $g$, 4°C for 10 min, to precipitate the mitochondrial fraction. Mitochondrial pellets were stored at −80°C until use.

The BNGE was carried out as described by Schägger and von Jagow (1991) with minor modifications. Briefly, mitochondrial lysates were prepared by incubating mitochondrial suspension in the ice-cold lysis buffer (50 mM Tris–HCl, pH 7.0, 750 mM aminocapoic acid, and 1.7% n-dodecyl-β-ᴅ-moltoside) for 10 min. The lysates were then clarified by centrifugation at 10,000 $g$, 4°C for 30 min. The protein concentrations were determined using the bicinchoninic acid assay reagents (Pierce). The gel loading mixtures were prepared by adding 10× loading buffer (750 mM aminocapoic acid and 3% Coomassie blue brilliant G-250) to the 60 µg of mitochondrial lysates. The BNGE was carried out by running the samples at 80 V/150 V on a 3–12 or 3%–16% gradient gel prepared from 41.6%, 100:1 acrylamide/bis-acrylamide stock solution and 3× gel buffer (150 mM Bis–Tris, pH 7.0, and 1.5 M aminocaproic acid) and using separate anode (50 mM Bis–Tris, pH 7.0) and cathode (15 mM Bis–Tris and 50 mM tricine with or without 0.02% Coomassie blue brilliant G-250) buffers. For Coomassie blue staining, the gels were fixed in a 50% methanol/10% acetic acid solution, stained with 0.1% Coomassie blue R-250 in the fixing solution, and destained with a 40% methanol/10% acetic acid solution. For Western blotting, the resolved protein complexes were transferred onto polyvinylidene fluoride membrane and then probed with either NDUFS3 antibody or the Total OXPHOS antibody cocktail. For the in-gel activity assay, Complex I activity was developed by incubating the gel in a solution containing 50 mM potassium phosphate buffer, pH 7.0, 0.1 mg/ml NADH, and 0.2 mg/ml nitrotetrazolium blue.

## Supplemental Information

## Acknowledgements

We sincerely thank Drs. Todd Leff, James Granneman, and Shijie Sheng for critical comments on the manuscript.

### Author Contributions

S Lucas: investigation.
G Chen: investigation.
S Aras: investigation and methodology.
J Wang: conceptualization, formal analysis, funding acquisition, investigation, methodology, and writing—original draft, review, and editing.

### Conflict of Interest Statement

The authors declare that they have no conflict of interest.

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
