## [Reviewer comments · Life Science Alliance]

Serine catabolism is essential to maintain mitochondrial respiration in mammalian cells

Stephanie Lucas, Guohua Chen, Siddhesh Aras and Jian Wang

DOI: 10.26508/lsa.201800036

Review timeline:	Submission Date:	22 February 2018
	1 st Editorial Decision:	22 March 2018
	1 st Revision Received:	30 April 2018
	2 nd Editorial Decision:	7 May 2018
	2 nd Revision Received:	7 May 2018
	Accepted:	8 May 2018

Report:

(Note: Letters and reports are not edited. The original formatting of letters and referee reports may not be reflected in this compilation.)

Please note that the manuscript was submitted on January 24th to an alliance journal and was 'scooping protected' as per *Life Science Alliance's* extended 'scooping protection': <http://www.life-science-alliance.org/editorial-policies#scoop>

1st Editorial Decision 22 March 2018

Thank you for submitting your manuscript entitled "Serine catabolism is essential to maintain mitochondrial respiration in mammalian cells" to Life Science Alliance. Your manuscript has now been reviewed by three referees whose comments are included below.

As you will see, while the referees overall appreciate the work and its potential significance, concerns were raised about the interpretation of some of the key experiments, as well as a need to cite and discuss appropriately recent papers that also examined SHMT2 function in mitochondrial homeostasis. If you feel that the manuscript can be modified according to the referees' suggestions, we would be happy to consider a revised manuscript for publication in Life Science Alliance.

In considering a revised manuscript, we suggest focusing on the following items:

1. Recent work reporting similar requirements for SHMT2 in supporting mitochondrial function (Mol Cell. 2018 Feb 15;69(4):610-621 and Nature. 2018 Feb 1;554(7690):128-132) need to be cited and discussed appropriately, particularly along the lines of comments 1 and 2 raised by Referee 2.
2. While the present study points to a defect in complex I assembly in the absence of SHMT2, recent studies have revealed instead an impairment in translation. Importantly, ribosome profiling was performed in support of the previously reported translation defect, but such analyses were not included in the current study (please see comment 2 from Referee 2). In light of this key difference, the manuscript text should be modified to soften the conclusion that translation is not defective in the absence of SHMT2, as previous work has shown this more definitively. The present results may still point toward defective complex I assembly as another contributing factor to mitochondrial dysfunction.
3. Reviewer 1 suggests a rescue experiment with formate to examine the effects on proliferation of Δ SHMT2 cells in galactose, which could be important to demonstrate that the observed restored

assembly of complex I is functionally significant.

4. Please add quantitative data for the western blots shown throughout the paper along with the number of replicates and appropriate statistical analyses.

REFEREE REPORTS

Reviewer #1 (Comments to the Authors (Required)):

Lucas et al. describe the effects of the inhibition of serine catabolism on the mitochondrial function. Their findings suggest "a novel regulatory link between SHMT2-mediated 1C metabolism and the maintenance of the mitochondrial respiratory chain in mammalian cells". These findings shed light on a previously unexplored and potentially relevant role of 1C metabolism in mitochondrial homeostasis. Despite the convincing biochemical data about the effects of the depletion of SHMT2 on mitochondrial function, the data presented are preliminary. Further analysis is required.

Specific comments:

1- The statement of increased glycolytic flux is based on lactate secretion. To verify this finding either glucose uptake or glycolytic rate (using radioactive tracers) should be measured.

2- Data for Δ SHMT1 need to be included throughout the experiments. Key data need to be recapitulated in at least one other cell line.

3- The authors correctly infer from the analysis that there are not downregulated genes in the Δ SHMT2. However, several genes appear to be upregulated. Do the authors further characterized these genes? Could they potentially have a role in the biological effect of SHMT2 depletion?

4- Error bars and statistics should be shown on all the graphs. The authors should always clarify the number of replicates and the statistical analysis performed.

5- What are the formate levels in Δ SHMT1 and Δ SHMT2 cells? What is the general activity of SHMT1 vs SHMT2 in the tested cell lines (can be measured using deuterated serine as described before)? Is proliferation upon formate addition rescued in Δ SHMT2 cells grown in galactose?

Reviewer #2 (Comments to the Authors (Required)):

The present manuscript confirms recent data showing that SHMT2 is required to support respiration by promoting Complex I levels. The manuscript includes compelling evidence confirming this main effect. It further duplicates recently published data showing that the change in mitochondrial complex levels occurs without changes in mitochondrial DNA abundance or transcripts. The paper in addition makes a few claims that could easily be misinterpreted to be in conflict with recently published data (Morscher et al), although the extent and quality of the present data is not sufficient to actually show this:

1. The present manuscript focuses exclusively on complex I, whereas the recently published data shows effects also, albeit weaker, on Complex IV and V. The subunits of those complexes that showed effect in the prior work were not measured here. The authors should make these measurements or explicitly indicate that they have failed to test for these previously described effects.

2. The present manuscript claims that there are no changes in translation based on one poor quality radiograph, with apparent errors in loading amount and running quality/alignment. The large body of recently published data showing that these effects arise due to translation needs to be directly confirmed (or not) before including data that claims to refute this mechanism, including improved radiographs AND also ribosome profiling data, which provided the most compelling evidence for translational impairment in Morscher et al.

3. Claims about an assembly mechanism are interesting but unwarranted until the translation mechanism can be disproved.

4. The most important deficiency of the prior manuscript is failure to cite the recently published data! While the present Journal is okay with confirmations, it is unacceptable to simultaneously mimic (down to certain aspects of the presentation) Morscher et al (published 1 month prior to this submission), while ignoring to cite it. This is a serious mistake that should not be repeated.

Reviewer #3 (Comments to the Authors (Required)):

Review of "Serine catabolism is essential to maintain mitochondrial respiration in mammalian cells" for Life Science Alliance.

Summary:

Lucas et al. study the role of SHMT1 and SHMT2 in energy metabolism. The authors show that knocking out SHMT2 in cultured cells leads to impaired respiratory chain function and decreased cell proliferation. They also found that impaired ATP production in these cells is due to downregulation of Complex I of the respiratory chain.

The study is well executed using appropriate methods and the major conclusions are supported by the data. However, most findings in the manuscript have already been reported in the literature and the reported findings are mostly confirmatory.

Specific comments:

The mechanistic basis of the observations in the manuscript has been studied earlier but the references to these papers are lacking. Especially important would be to acknowledge the work of Morscher et al (Nature 2018) and Minton et al (Mol Cell 2018).

Figure 1A:

The authors conclude that "In addition, the levels of key glycolytic enzymes including pyruvate kinase M (PKM) and lactate dehydrogenase (LDH) were elevated in the SHMT2 cells (Fig 1A)." This is not completely supported by the data due to semi-quantitative nature of western blotting. The induction seems to be small and this raises the question if the changes are biologically significant and reproducible. Therefore, the conclusion should be modified or more evidence provided in support of the current conclusion.

1st Revision – authors' response

30 April 2018

Responses to the comments of **Reviewer #1**

"Lucas et al. describe the effects of the inhibition of serine catabolism on the mitochondrial function. Their findings suggest "a novel regulatory link between SHMT2-mediated IC metabolism and the maintenance of the mitochondrial respiratory chain in mammalian cells". These findings shed light on a previously unexplored and potentially relevant role of IC metabolism in mitochondrial homeostasis. Despite the convincing biochemical data about the effects of the depletion of SHMT2 on mitochondrial function, the data presented are preliminary. Further analysis is required."

1. "The statement of increased glycolytic flux is based on lactate secretion. To verify this finding either glucose uptake or glycolytic rate (using radioactive tracers) should be measured."

Our response: We sincerely thank the Reviewer's suggestion on how to further define the glycolytic metabolism. We will apply these approaches to our future studies. In the current study, we want to follow the Editor's suggestion of focusing on the aspects of mitochondrial respiration.

2. "Data for Δ SHMT1 need to be included throughout the experiments. Key data need to be recapitulated in at least one other cell line."

Our response: In our original manuscript, we have parallelly examined the relatively-broad effects associated with SHMT1 and SHMT2 deficiency. Through these studies, we have identified a unique dependence of mitochondrial respiration on SHMT2. Also because of our limited research resource,

we feel that further characterization of DSHMT1 cells may not be necessary for the part of our study that investigates how SHMT2 modulates mitochondrial respiration. In our original manuscript, we have created targeted deletion of SHMT2 in AML12 cells, in addition to that in 293A cells. We have observed a consistent respiratory defect consequent to SHMT2 deletion in both of the cell lines. We hope these results in our manuscript will help to address the Reviewer's concern.

3. "The authors correctly infer from the analysis that there are not downregulated genes in the Δ SHMT2. However, several genes appear to be upregulated. Do the authors further characterized these genes? Could they potentially have a role in the biological effect of SHMT2 depletion?"

Our response: We agree that the elevated expression of some OXPHOS genes in response to SHMT2 loss is indeed interesting and potentially important. We think that this could be a compensatory response consequent to the respiratory impairment in the absence of SHMT2. In the current study, we have focused on the investigation of the cause for the respiratory defect in the absence of SHMT2. We hope the Reviewer will agree that we can follow up on this interesting observation in future studies.

4. "Error bars and statistics should be shown on all the graphs. The authors should always clarify the number of replicates and the statistical analysis performed."

Our response: We agree with the Reviewer's comment. According to the reviewer's suggestion, we included error bars in the revised Figure 3D, as well as the information of experimental replicates and statistical methods in the revised figure legends.

5. "What are the formate levels in Δ SHMT1 and Δ SHMT2 cells? What is the general activity of SHMT1 vs SHMT2 in the tested cell lines (can be measured using deuterated serine as described before)? Is proliferation upon formate addition rescued in Δ SHMT2 cells grown in galactose?"

Our response: The regulations of the serine metabolic flux as well as the one-carbon metabolite level by SHMT1 and SHMT2 have been extensively characterized in HEK293 cells, previously (Bao *et al*, eLIFE 2016 5:e10575, Ducker *et al*, Cell Metab 2016 23:1140-1153). According to the Reviewer's suggestion, we performed the rescue experiment to test whether formate supplementation impacts on the proliferation of DSHMT2 HEK293 cells in galactose. We observed that formate completely restored the proliferation of the DSHMT2 cells in galactose. Thanks to the Reviewer's suggestion, we believe this result provides an important piece of evidence illustrating the critical role of SHMT2-mediated one-carbon provision in supporting mitochondrial respiration. This result is presented in the revised Figure 5E.

Responses to the comments of **Reviewer #2**

"The present manuscript confirms recent data showing that SHMT2 is required to support respiration by promoting Complex I levels. The manuscript includes compelling evidence confirming this main effect. It further duplicates recently published data showing that the change in mitochondrial complex levels occurs without changes in mitochondrial DNA abundance or transcripts. The paper in addition makes a few claims that could easily be misinterpreted to be in conflict with recently published data (Morscher *et al*), although the extent and quality of the present data is not sufficient to actually show this:"

1. "The present manuscript focuses exclusively on complex I, whereas the recently published data shows effects also, albeit weaker, on Complex IV and V. The subunits of those complexes that showed effect in the prior work were not measured here. The authors should make these measurements or explicitly indicate that they have failed to test for these previously described effects."

Our response: We are sorry we didn't make our points clear in our manuscript. In fact, we measured the expression of all five individual respiratory complexes at both the single-subunit (Figure 3A) and mature-complex levels (Figures 4A and 5C). We found that SHMT2 deletion selectively depleted Complex I, and had a negligible effect on the other four complexes, at both the single-subunit and mature-complex levels. Because the status of the mature complexes has not been reported by the previous reports (Morscher *et al*, Nature 2018 554:128-132, Minton *et al* Mol Cell 2018 69:610-621), we believe the results to be reported in our manuscript will be an important addition to the understanding of the regulation of respiratory function by SHMT2. For improving clarity, we revised the result sections related in the text.

2. "The present manuscript claims that there are no changes in translation based on one poor quality radiograph, with apparent errors in loading amount and running quality/alignment. The large body of recently published data showing that these effects arise due to translation needs to be directly confirmed (or not) before including data that claims to refute this mechanism, including

improved radiographs AND also ribosome profiling data, which provided the most compelling evidence for translational impairment in Morscher *et al.*”

Our response: We agree with the reviewer’s comment and we are sorry for the data quality of the radiograph. We optimized the experimental condition, and repeated the 35S-Met/Cys pulse labeling experiment. To further enhance accuracy, we analyzed each cell line in duplicate, and verified gel loading by Coomassie staining of total protein. Consistent with our previous result, we were unable to observe an appreciable alteration in the biosynthesis of any mitochondria- encoded OXPHOS proteins in the SHMT2-deficient cells. These results are presented in the revised Figure 3E. It is noteworthy that the recent reports by Morscher *et al* (Nature 2018) and Minton *et al* (Mol Cell 2018) have reported clearly differential translation responses to SHMT2 deletion in different cell types. We think that the unaltered mitochondrial protein biosynthesis in our cell model could also represent a kind of cell-type-specific response. Regardless, we revised our text to acknowledge that more sophisticated approach such as ribosomal profiling will help to detect more subtle translational change, if any, in our cell model.

3. “Claims about an assembly mechanism are interesting but unwarranted until the translation mechanism can be disproved.”

Our response: We agree with the Reviewer’s comment. By far, our results pointed to a lack of translational defect in our cell model. We realize that previous studies have reported the translational impairment in the different cell types, and that more sophisticated approach such as ribosomal profiling could detect more subtle translational change in our cell model. We softened our conclusion and stated in the revised text that complex assembly is another contributing mechanism in addition to translation control.

4. “The most important deficiency of the prior manuscript is failure to cite the recently published data! While the present Journal is okay with confirmations, it is unacceptable to simultaneously mimic (down to certain aspects of the presentation) Morscher *et al* (published 1 month prior to this submission), while ignoring to cite it. This is a serious mistake that should not be repeated.”

Our response: According to the reviewer’s comment, we cited and discussed the recent reports by Morscher *et al* and Minto *et al* in our revision.

Responses to the comments of **Reviewer #3**

“Lucas *et al.* study the role of SHMT1 and SHMT2 in energy metabolism. The authors show that knocking out SHMT2 in cultured cells leads to impaired respiratory chain function and decreased cell proliferation. They also found that impaired ATP production in these cells is due to downregulation of Complex I of the respiratory chain.

The study is well executed using appropriate methods and the major conclusions are supported by the data. However, most findings in the manuscript have already been reported in the literature and the reported findings are mostly confirmatory.

Specific comments:”

“The mechanistic basis of the observations in the manuscript has been studied earlier but the references to these papers are lacking. Especially important would be to acknowledge the work of Morscher *et al* (Nature 2018) and Minton *et al* (Mol Cell 2018).”

Our responses: According to the reviewer’s comment, we cited and discussed the recent reports by Morscher *et al* and Minto *et al* in our revision.

“Figure 1A:

The authors conclude that “In addition, the levels of key glycolytic enzymes including pyruvate kinase M (PKM) and lactate dehydrogenase (LDH) were elevated in the SHMT2 cells (Fig 1A).” This is not completely supported by the data due to semi-quantitative nature of western blotting. The induction seems to be small and this raises the question if the changes are biologically significant and reproducible. Therefore, the conclusion should be modified or more evidence provided in support of the current conclusion.”

Our response: We agree with the Reviewer’s comment. We quantified the Western blot results by densitometry scanning. The quantification revealed a modest and significant increase of PKM and LDH expression in the SHMT2-deficient cells. We modified our conclusion and stated in the revised text that loss of SHMT2 modestly enhances PKM and LDH expression.

Thank you for submitting your revised manuscript entitled "Serine catabolism is essential to maintain mitochondrial respiration in mammalian cells". Your manuscript has been re-assessed by two of the original referees. As you will see, the reviewers appreciate the introduced changes, and we would thus be happy to publish your paper in Life Science Alliance pending final text revisions necessary to address the valid comments made by reviewer #2 (see below).

REFEREE REPORTS

Reviewer #1 (Comments to the Authors (Required)):

The revised version of the manuscript acknowledges prior work, has a softened statement on translational defects, includes a supportive formate rescue experiment in galactose, and has statistics as well as WB quantification. In my opinion the authors sufficiently addressed the requests highlighted by the reviewers and prioritized by the editor.

Reviewer #2 (Comments to the Authors (Required)):

This improved revision still needs to acknowledge in the abstract the current state of the field, changing "However, little is known about the biological effects of serine catabolism on cellular energy production. Using CRISPR-Cas9 gene editing, we demonstrate that the mitochondrial SHMT enzyme," to something like "Recently it has been shown that serine catabolism is required for efficient respiration. Using CRISPR-Cas9 gene editing, we confirm that the mitochondrial respiration..."

The authors should also acknowledge that the Morscher paper explicitly showed that Complex I is most strongly affected. Intriguingly, this matches with the codon usage associated with the ribosome stalling. Perhaps the ribosome stalling impairs efficient complex assembly even if translation proceeds to completion?

2nd Revision – authors' response

7 May 2018

In response to the comments of the Reviewer #2, we revised the text in Abstract and Discussion in our manuscript. Revisions in the text are shown using blue font for additions and red strikethrough font for deletions. The new revisions are highlighted in yellow. We hope you will agree that the revised manuscript addresses the concerns of the Reviewer, and that our manuscript is suitable for publication in **Life Science Alliance**.

3rd Editorial Decision

8 May 2018

Thank you for submitting your Research Article entitled "Serine catabolism is essential to maintain mitochondrial respiration in mammalian cells". It is a pleasure to let you know that your manuscript is now accepted for publication in Life Science Alliance. Congratulations on this interesting work.